

# The relationship between distal trunk morphology and object grasping in the African savannah elephant (*Loxodonta africana*)

Julie Soppelsa[1], Emmanuelle Pouydebat[1], Maëlle Lefeuvre[2], Baptiste Mulot[3], Céline Houssin[4] and Raphaël Cornette[4]

[1] Adaptive Mechanisms and Evolution (MECADEV), Centre national de la recherche scientifique/Muséum national d'Histoire naturelle, Paris, France
[2] Institute of Environmental Sciences, Faculty of Biology, Jagiellonian University Cracow, Cracow, Poland
[3] Zooparc de Beauval & Beauval Nature, Saint-Aignan, France
[4] Institut de Systématique, Evolution, Biodiversité (ISYEB), Centre national de la recherche scientifique/Muséum national d'Histoire naturelle, SU, EPHE, UA, Paris, France

Corresponding author
Julie Soppelsa,
julie.soppelsa@edu.mnhn.fr

## ABSTRACT

**Background**. During reach-to-grasp movements, the human hand is preshaped depending on the properties of the object. Preshaping may result from learning, morphology, or motor control variability and can confer a selective advantage on that individual or species. This preshaping ability is known in several mammals (*i.e.,* primates, carnivores and rodents). However, apart from the tongue preshaping of lizards and chameleons, little is known about preshaping of other grasping appendages. In particular, the elephant trunk, a muscular hydrostat, has impressive grasping skills and thus is commonly called a hand. Data on elephant trunk grasping strategies are scarce, and nothing is known about whether elephants preshape their trunk tip according to the properties of their food.

**Methods**. To determine the influence of food sizes and shapes on the form of the trunk tip, we investigated the morphology of the distal part of the trunk during grasping movements. The influence of food item form on trunk tip shape was quantified in six female African savannah elephants (*Loxodonta africana*). Three food item types were presented to the elephants (elongated, flat, and cubic), as well as three different sizes of cubic items. A total of $107 \pm 10$ grips per individual were video recorded, and the related trunk tip shapes were recorded with a 2D geometric morphometric approach.

**Results**. Half of the individuals adjusted the shape of the distal part of their trunk according to the object type. Of the three elephants that did not preshape their trunk tip, one was blind and another was subadult.

**Discussion and perspectives**. We found that elephants preshaped their trunk tip, similar to the preshaping of other species' hands or paws during reach-to-grasp movements. This preshaping may be influenced by visual feedback and individual learning. To confirm these results, this study could be replicated with a larger sample of elephants.

## INTRODUCTION

Grasping is an essential motion that is widespread among many tetrapod species and plays a fundamental role in feeding, social interactions and locomotion (*Pouydebat et al., in press*; *Sustaita et al., 2013*). Various grasping techniques can be performed with mouths, hands, paws, feet, or trunks (*Christel, 1993*; *Sustaita et al., 2013*). For example, elephants often grasp each other's trunks during greeting ceremonies, bonding ceremonies and social play (*Poole & Granli, 2011*).

Hand grasping, in particular, has been widely studied among primates, including humans, because of their individual fingers and ability to perform complex grasping and manipulation tasks (*Napier, 1956*; *Christel, 1993*; *Jones-Engel & Bard, 1996*; *Pouydebat et al., 2011*). In primates, objects, including tools and food, are grasped according to their physical properties, such as size, shape, consistency, and mobility, and depending on the action planned (*Ansuini et al., 2006*; *Ansuini et al., 2008*; *Chieffi & Gentilucci, 1993*; *Jeannerod, 1984*; *Pouydebat et al., 2009*; *Santello, 2002*; *Santello, Flanders & Soechting, 2002*).

Many primates preshape their hands according to food properties during grasping, including chimpanzees (*Jones-Engel & Bard, 1996*; *Pouydebat et al., 2009*), macaques (*MacFarlane & Graziano, 2009*; *Sartori et al., 2013*) and strepsirrhines (*Peckre et al., 2019*; *Reghem et al., 2011*; *Toussaint et al., 2013*; *Toussaint et al., 2015*). However, when reaching to grasp the same object, *i.e.*, when the hand moves towards the same object to be grasped, the general shape of macaque and human hands differ (*Roy et al., 2000*). In species besides primates, such as rats (*Whishaw & Coles, 1996*; *Sacrey, Alaverdashvili & Whishaw, 2009*) and frogs (*Anzeraey et al., 2017*), food properties also affect grasping strategies.

More precisely, during reach-to-grasp movements, the human hand gradually adjusts (*Santello & Soechting, 1998*; *Mason, Gomez & Ebner, 2001*) to accommodate the object contour (*Jeannerod, 1984*; *Pellegrino, Klatzky & McCloskey, 1989*; *Chieffi & Gentilucci, 1993*; *Supuk, Kodek & Bajd, 2005*). Similar behaviour has been observed for other primate species, such as macaques (*Roy et al., 2000*; *Sartori et al., 2013*) and capuchins (*Christel & Fragaszy, 2000*). The object affordances (*i.e.*, the properties of an object; *Sartori, Straulino & Castiello, 2011*) evoke its use and function during grasping.

Preshaping enables increased grasping success (*Sartori, Straulino & Castiello, 2011*) and can thus serve as a selective advantage for species or individuals. Preshaping may result from learning or variability in appendage morphology or motor control.

While hand preshaping during reach-to-grasp movements in primates is well established, data on other species is scarce. Only rats (*Whishaw, Dringenberg & Pellis, 1992*; *Sacrey, Alaverdashvili & Whishaw, 2009*) and pigeons (*Bermejo & Zeigler, 1989*) have been studied in terms of their ability to preshape their paw or beak when grasping an item. Despite the enormous variability in trunk utilisation for grasping objects in elephants (*Lefeuvre et al., 2020*; *Poole & Granli, 2011*; *Wu et al., 2018*; *Yasui & Idani, 2017*), studies have not yet investigate the potential for elephant trunk preshaping during reach-to-grasp movements.

Elephant trunks are especially interesting appendages due to this grasping variability. Elephants use their trunks in many situations, such as grasping food, congeners or other items in their environment (*Lefeuvre et al., 2020*; *Lee & Moss, 2014*; *Poole & Granli, 2011*;

*Riyas Ahamed, 2015*; *Yasui & Idani, 2017*). The trunk is a muscular hydrostat that does not have hard skeletal elements, similar to cephalopod tentacles and the tongues of many vertebrates; thus, it can bend in any direction, elongate, shorten and lift heavy masses (*Kier & Smith, 1985*; *Wilson et al., 1991*). Grasping techniques, mainly including the morphological and kinematic aspects, are well studied in muscular hydrostats (*Grasso, 2008*; *Ritter & Nishikawa, 1995*). Elephant grasping has also received increasing attention, especially the techniques that involve the distal part of the trunk (*Dagenais et al., 2021*; *Schulz et al., 2021*). Although the most commonly used grasping technique in muscular hydrostats is coiling around the item, elephant trunk tips can also engage in pinch grasping (*Rasmussen & Munger, 1996*; *Wu et al., 2018*) as can chameleon tongues (*Herrel et al., 2000*). Indeed, African elephant can perform precise pinch grasping movements. According to the IUCN, there are two distinct African elephant species: the African savannah elephant (*Loxodonta africana*) and the African forest elephant (*Loxodonta cyclotis*) (*Hart et al., 2021*). Unlike Asian elephants, which have only one finger at the top of their trunk tip, African elephants can pinch items with high precision between two fingers (one at the bottom and one at the top of the trunk tip; *Hoffmann, Montag & Dominy, 2004*). The grasping capacities of the African elephant trunk tip are sometimes compared to those of appendages of other species, such as the human hand (*Onodera & Philip Hicks, 1999*; *Hoffmann, Montag & Dominy, 2004*). Despite these intriguing grasping capacities, no studies have examined the potential for preshaping in muscular hydrostats to date; thus, none have investigated the potential for preshaping of trunk tips.

In this study, we aimed to understand grasping behaviours in African savannah elephants. As grasping capacities are impacted by individual variability in the shape of the appendage, we first investigated whether there were interindividual differences in the trunk tip shape. Then, we determined whether elephants preshape their trunk tips during reach-to-grasp movements toward food items and whether this potential preshaping influences grasping success. Last, we examined whether grasping success increased as the trials progressed and whether all elephants exhibited similar trunk grasping shapes for a given item by the end of the trials.

We predicted that each elephant has a unique distal trunk shape and that preshaping of the trunk tip would occur during reach-to-grasp movements toward food items. We also expected that preshaping of the trunk tip would increase grasping success and that over time an elephant's grasping movements would become more efficient. Finally, we predicted that all individuals would gradually converge toward the same distal trunk shape when reaching to grasp the same object. To test these hypotheses, we quantified and visualised (with geometric morphometric tools) the morphology of the distal part of the trunk during reach-to-grasp movements toward food items for six female African savannah elephants (*Loxodonta africana*; Blumenbach, 1797).

## MATERIALS & METHODS

### Subjects and housing

Data presented in this study were collected on six African savannah elephants (*Loxodonta africana*) from the ZooParc of Beauval in Loir-et-Cher, France: Ashanti, Juba, Marjorie,

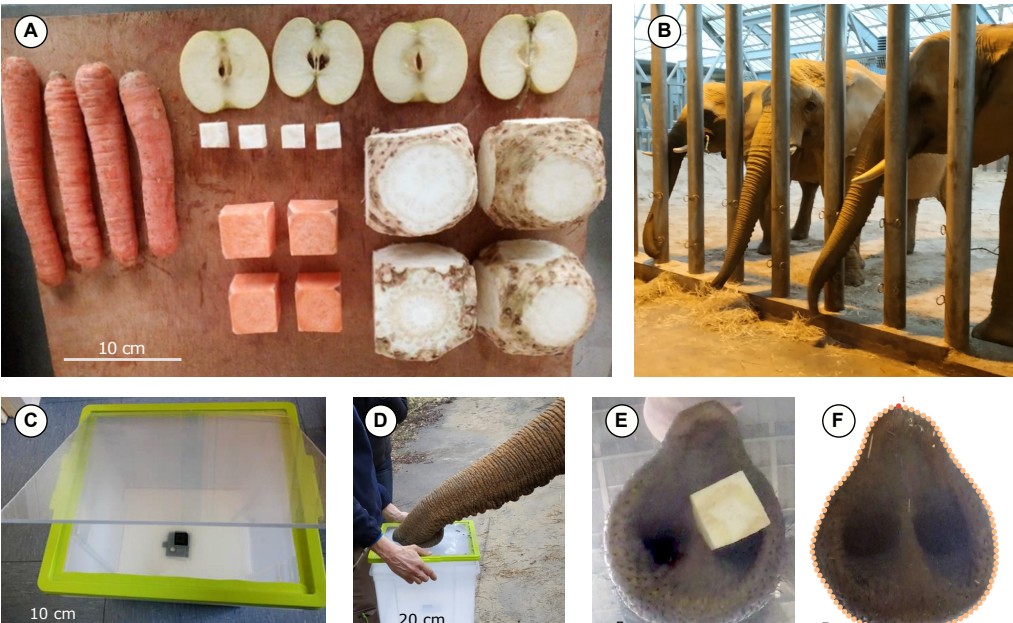

**Figure 1** **The materials and methods used for the experiments.** (A) Example of food items used for the experiment. Here we presented to the elephant five type of item: carrot, flat apple slice, two centimeter side-length celery cube, four centimeter side-length sweet potato cube and eight centimeter side-length celery cube, with a repeatability of four for each. (B) Three of the six African savannah elephants (*Loxodonta africana*) of the ZooParc of Beauval. (C) The double box system used to film the experiment. It is a double transparent box inlay in one another, the first one contains the camera and polystyrene to maintain it, and the second one had a hole at its bottom for the lens of the camera. (D) The items were put down on the transparent top of the second box and carried in front of the animal. (E) Example of a screenshot of a four centimeter side-length food cube just before grasp, in order to take the print of the distal part of the trunk. (F) Example of a trunk tip contouring during a grasp. The red dot corresponds to the anatomical landmark at the top of the trunk tip, and the 100 orange dots correspond to the sliding semi-landmarks around the outlines of the distal part.

M'Kali, N'Dala and Tana (Fig. 1B). Five of these elephants were approximately 32 years old at the beginning of the experiment; Ashanti was a 16-year-old subadult. Comparisons between the adults and the subadult allowed us to examine the role of experience in trunk preshaping behaviours. All elephants except Ashanti were wild-born. They lived in two groups, each of which had a specific dominance hierarchy. One of the females, N'Dala, had been blind for a year by the time the study commenced. Thus, comparisons between sighted and blind elephants allowed us to assess the importance of visual input for grasping behaviours. The elephant characteristics can be found in Table 1.

The elephants were housed together but were isolated during the experiment. They were tested during their weekly training sessions. Every morning, one elephant was trained to facilitate any future medical interventions; this corresponded to one training session per elephant per week. The elephants were already experienced with grasping food items during these weekly training sessions; thus, they were already experienced with the movements required for the experiment. The running order of these training sessions changed each

**Table 1  Subjects characteristics.** These characteristics include the names of the elephants, their sex, age, hierarchical rank, origin and physical particularities.

| Name | Sex | Age (year) | Hierarchical rank | Origins | Physical particularities |
|------|-----|-----------|-------------------|---------|--------------------------|
| Ashanti | Female | 16 | Medium rank | Knowsley Safari Park (UK) | Smallest elephant |
| Juba | Female | 32 | High rank | Zimbabwe | – |
| Marjorie | Female | 33 | High rank | Zimbabwe | Largest elephant |
| M'Kali | Female | 30 | Low rank | Namibia | Tall |
| N'Dala | Female | 30 | Medium rank | Namibia | Blind, trunk tip deformity |
| Tana | Female | 32 | Medium rank | Zimbabwe | – |

**Table 2  Dimensions of the food items grasped by the elephants.** Dimensions in millimetres.

| Food item | Dimensions (mm) |
|-----------|-----------------|
| Apple slice | 75 (diameter) |
| Carrot | 160 (length) * 25 (diameter) |
| Small cube | 20*20*20 |
| Medium cube | 40*40*40 |
| Large cube | 80*80*80 |

week. Thus, in total, we conducted five to six experimentation sessions per elephant over six weeks in spring 2019.

## Ethical note

As the data were collected by keepers during the weekly elephant training sessions, only standard elephant-keeper interactions occurred, in accordance with zoo security regulations. All procedures were conducted in accordance with the relevant CNRS guidelines and European Union regulations (Directive 2010/63/EU).

## Object grasping

The food items used in the experiments were found in the typical diet of captive Beauval elephants and included carrots, apples and celery (*Ullrey, Crissey & Hintz, 1997*; *Olson, 2004*). Three food item types were presented to the elephants: elongated objects with a constant diameter (carrots), flat objects (apple slices, cut horizontally), and cubic objects of three sizes. The objects consisted of diverse vegetables (according to availability), and the three cubic sizes included small cubes (two centimetres per side), medium cubes (four centimetres per side) and large cubes (eight centimetres per side). Food item dimensions are presented in Table 2. The small cube size was determined by elephant participation: smaller cubes (*i.e.,* one centimetre per side) made the elephants impatient, and experiments with this size were inconclusive. The large cube size was determined according to the maximum size of the largest vegetables (*e.g.,* beetroots and celery). Three to six items of each type were provided during each session (Fig. 1A), depending on vegetable availability, except for the large cubes, which had low availability.

## Recordings of grasping movements

To determine the influence of food item characteristics on the trunk tip form during grasping behaviours, the items were laid out on a double transparent box. The bottom of the first box contained the camera, set in polystyrene, and a second box, with a hole at its bottom for the lens of the camera, was placed over the first box. The top of the second box was a transparent lid (Fig. 1C) where the items were individually placed. This box-and-camera system was placed on the ground in front of the animal by an animal keeper (Fig. 1D). Therefore, we were able to film all grasping behaviours from below. An example film of a grasping behaviour can be found in the Supplementary Material.

We recorded the reach-to-grasp movements of each individual with a digital video camera (GoPro7). For each grasp, we took pictures of the distal part of the trunk. We captured these images from the video frames when the elephant first touched the box and when the trunk was flat against the box (Fig. 1E). Other grasping techniques, such as coiling, were removed from the analysis. For each grasping attempt, we also noted when the trunk hit the box near the food but missed the food item. As these grasping attempts require a readjustment of the trunk towards the item, they are directly correlated with the time spent on a grasp. The elephants almost never dropped food items but took longer to grasp them after missing. Therefore, we termed these grasping attempts "failure before a grasp" to differentiate them.

## Geometric morphometrics and statistical analysis

To describe the shape of the tip of the trunk, we used a 2D geometric morphometric (GM) approach (*Rohlf & Marcus, 1993*; *Adams, Rohlf & Slice, 2004*; *Gunz & Mitteroecker, 2013*). For each image of the trunk tip, the trunk tip was outlined with tpsDIG2 software, version 2.32 (*Rohlf, 2016*). To obtain these outlines, we placed an anatomical landmark on the top of each trunk tip and placed 100 sliding semilandmarks on the curves (*Gunz & Mitteroecker, 2013*) to outline the distal part of the trunk (Fig. 1F). These semilandmarks defined the borders of the trunk tip. Every characteristic (the subject, food item, and number of failures) that might influence the trunk tip contour was annotated. The film in the Supplementary Material shows a part of this methodology (video recording of the grasp as well as the landmark and semilandmarks on the trunk tip).

We performed a Generalised Procrustes Analysis (GPA) (*Rohlf & Slice, 1990*), using the "gpagen" function of the package "geomorph" version 3.2.1 (*Adams, Collyer & Kaliontzopoulou, 2018*) in R version 3.6.1 (*R Core Team, 2018*). This GPA aligned the different distal trunk shapes with a Procrustes superposition to remove the effects of translation, rotation, and scaling. We visualised the GPA using the "plotTangentSpace" function and retained 90% of shape variability after reducing the dimensionality of the variables with a principal components analysis (PCA) (*Baylac & Frieß, 2005*). The PCA components were used as shape parameters in the following analyses. To detect significant differences in the shape of the trunk according to the object properties, individual, or number of grasping failures, we performed a MANOVA using the "manova" function in R. A MANOVA was also used to detect significant differences in trunk tip shape according to the object properties for each individual.

To optimise and visualise individual differences in trunk tip shapes, canonical variable analysis (CVA) was performed using the "CVA" function in the package "Morpho" version 2.8 (*Schlager, 2017*). This CVA was applied to only five of the six elephants because one elephant, N'Dala, had a deformed trunk tip caused by a previous injury. Her trunk tip thus differed from those of the other five. To explore intact trunk tip shapes and obtain the most representative variability between individuals, N'Dala was excluded from this analysis. All other analyses included all six elephants.

A CVA was also performed for each individual to calculate the Mahalanobis distances between food items. These Mahalanobis distance plots revealed the similarity between trunk shapes for grasping different items. The Mahalanobis distances were calculated with the "hclust" function in R, plotted in an "unrooted" form and then paired with deformation vector fields and deformation isolines of the trunk tip shapes for each object. These thin plate spline (TPS) deformations showed the difference in shape between the mean trunk tip shape for each individual and the mean trunk tip form per object for each individual. To obtain these TPS deformations, every dot of the mean distal form outline was slid to the position of the mean distal form per object. A deformation grid was thus obtained, and the intensity and direction of the deformations were indicated by smooth deformation isolines and deformation vector fields. The deformation isolines indicated the location of the strongest differences in trunk tip form, and the deformation vector fields showed the direction of these differences. These TPS visualisations were performed with the functions "tps_arr" and "tps_iso" in the "Momocs" package (version 1.3.2) in R (*Klingenberg, 2013*).

The mean number of grasping failures per individual and per object for all trials were also calculated. Linear regressions were performed to determine the change in the number of failures over time for each individual and each object. The number of failure are shown along all the grasping attempts (all objects included). Finally, a GPA of the trunk tip shapes of all individuals for each object was performed and paired with trajectory plots. These plots were obtained in the Procrustes space by joining for each individual the dots corresponding to the outlines of the first and last grasping movement. We could therefore examine the trajectories of the distal trunk shapes during grasping to see if different individuals's trunk tips converged on similar shapes for the same object.

## RESULTS

We obtained a total of 639 grasps from 03:03 h of video from all studied elephants. Of these grasps, trunk tip contouring could be obtained from 491. In the remaining 148 grasps, the trunk either did not lay flat on the box lid or the object or side of the box obscured a part of the trunk tip. The distribution of the 491 grasps among each elephant and food item can be found in Table 3. The number of grasps were not equivalent for each individual and item type. These differences can be explained by the 148 grasps that were removed and by variation in the availability of vegetables in each session. Similarly, the low number of large-cube grasps is explained by the low availability of vegetables to create this item. Moreover, in the 491 grasps included in the final analysis, M'Kali only interacted once with

**Table 3 Number of grasping attempts per individual and per food item.** All these grasping were included in the analyses.

| Number of grasping attempts | | Individual | | | | | |
|---|---|---|---|---|---|---|---|
| | | *Ashanti* | *Juba* | *Marge* | *M'Kali* | *N'Dala* | *Tana* |
| Food item | Apple slice | 13 | 20 | 21 | 18 | 27 | 26 |
| | Carrot | 14 | 23 | 15 | 18 | 18 | 17 |
| | Small cube | 19 | 25 | 23 | 20 | 20 | 20 |
| | Medium cube | 15 | 19 | 15 | 20 | 20 | 19 |
| | Large cube | 4 | 6 | 6 | 1 | 5 | 4 |
| **Total grasping attempts** | | 65 | 93 | 80 | 77 | 90 | 86 |

**Table 4 Results of the variance analyses MANOVA performed on all the individuals trunk tip shapes just before a grasp and some variables.**

| All elephants trunk tip shapes according to | Pillai's Trace | F | df | *P*-value |
|---|---|---|---|---|
| Individuals | 2.67 | 93.12 | 5 | 2.2e-16*** |
| Objects | 0.10 | 1.79 | 5 | 0.0051** |
| Individuals * objects | 0.35 | 1.23 | 20 | 0.0363* |
| Number of failures | 0.12 | 1.10 | 9 | 0.2873 |

a large cube; the other interactions between M'Kali and the large cubes occurred in the 148 grasps that were removed.

Table 4 shows the results of the MANOVA on the trunk tip shape just before a grasp (the preshape) and the individual, object and number of failures variables. The trunk tip shape varied according to these variables. First, the distal trunk shape varied significantly by individual and by object ($p < 0.05$). The significant association between the distal trunk shape and the individual elephants supports the hypothesis of unique distal trunk shape. The significant association between the distal trunk shape and the type of object supports the hypothesis of trunk tip preshaping during a reach-to-grasp movement toward a food item. Thus, the trunk tip shape corresponded to the individual and the type of object. Moreover, the trunk tip shapes for each object were also dependent on the individual elephant. However, there was not a significant association ($p > 0.05$) between the distal trunk shape and the number of failures. Therefore, in this experiment, we were not able to establish a link between distal trunk shape and the number of failures before a successful grasp.

## Individual differences

As seen in Table 4, trunk tip shapes differed according to the individuals. Interindividual variability in the distal trunk shape during food item grasping is shown in Fig. 2. This CVA of the elephant trunk tip shape during food item grasping included only five of the six elephants because one elephant, N'Dala, had a deformed trunk tip; therefore, she was not included in this analysis. Each point in Fig. 2 represents a grasp, each colour represents an individual, and the shape at the end of each axis represents the shape of the maximum (red) and minimum (blue) trunk tip.

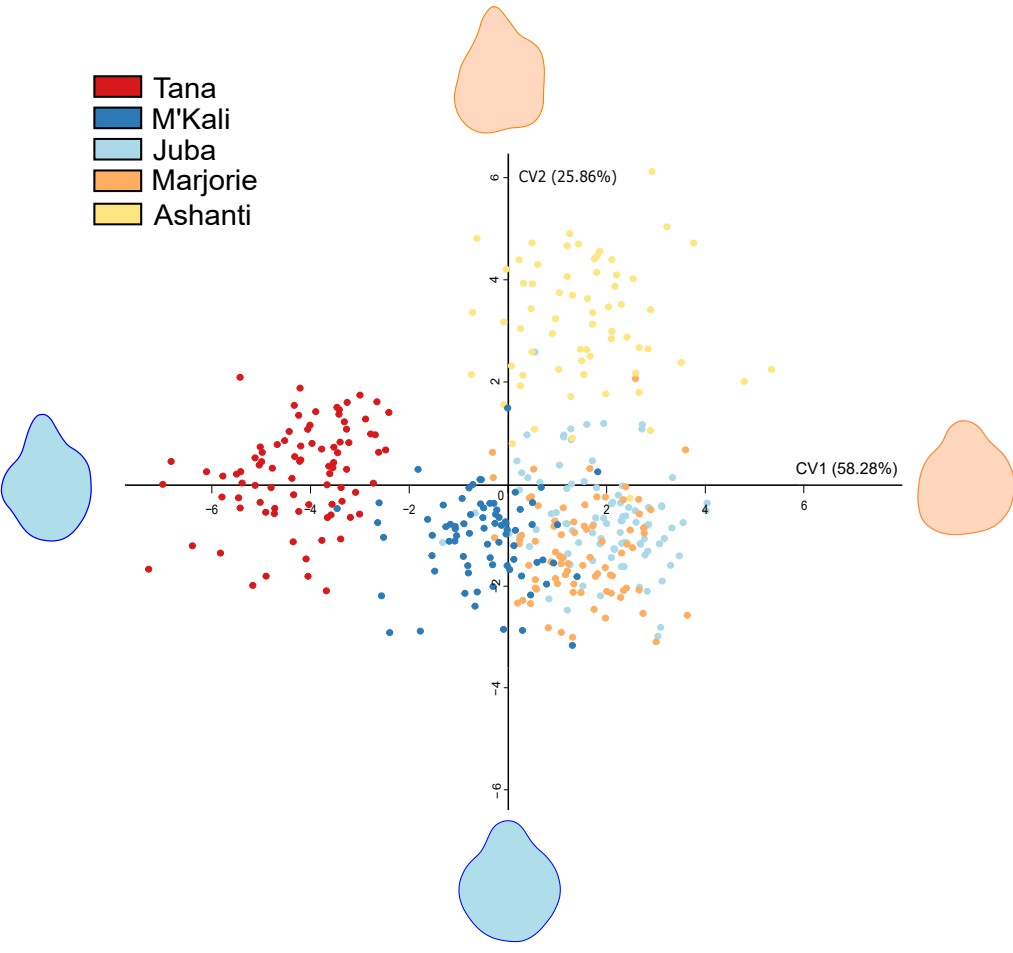

**Figure 2** **Canonical Variate Analysis visualisation of five elephant trunk tip shapes during food grasp.**
Each point represents a grasp, each colour an individual, and each end of the axis the maximum (red) and
minimum (blue) distal parts shapes.

In Fig. 2, all distal trunk shapes are distinctly grouped by individual. Although Juba's
(light blue) and Marjorie's points (orange), are grouped together, their distal trunk shapes
were still very different from each other. There were pronounced individual differences in
the distal trunk shape between elephants.

## Trunk shapes and object characteristics

Table 4 shows that the trunk tip shape varied according to the type of object grasped and
the individual elephant. Some of the studied elephants preshaped their trunk tip according
to the object properties. The results of the MANOVA on the trunk tip shape just before a
grasp, performed per individual and according to the type of object, are shown in Table 5.
The trunk tip shape significantly differed according to the type of object for M'Kali,
Juba and Marjorie; thus, there was a significant effect of object properties on trunk tip
preshaping. The three other elephants, N'Dala (the blind elephant), Ashanti (the subadult)
and Tana, did not preshape their trunk tip according to the food item properties. Thus,

**Table 5  Results of the variance analyses MANOVA performed per individuals on the trunk tip shapes just before a grasp according to objects.**

| Trunk tip shapes according to objects per individuals | Pillai's Trace | F | df | *P*-value |
|---|---|---|---|---|
| M'Kali | 0.49 | 2.29 | 3 | 0.0028** |
| Juba | 0.50 | 1.76 | 4 | 0.0113* |
| Marjorie | 0.34 | 1.79 | 4 | 0.0311* |
| N'Dala | 0.48 | 1.22 | 4 | 0.1848 |
| Ashanti | 0.51 | 1.20 | 4 | 0.2289 |
| Tana | 0.28 | 0.85 | 4 | 0.6762 |

only half of the sampled individuals had specific tip trunk shapes associated with object properties.

Specifically, Figs. 3–5 display the trunk tip shapes of M'Kali (Fig. 3), Juba (Fig. 4) and Marjorie (Fig. 5) and their variance according to the object properties. As M'Kali only interacted with the large cube once in the analysed grasps, her trunk tip shape associated with the large cube is not included in the figure. These three Mahalanobis distance plots paired with thin plate spline deformation visualisations represent the three individuals who preshape their trunks according to the object properties. The TPS deformations are the differences between the mean distal trunk shape for each object and the general mean shape, for each individual. The isoline visualisations show the location of these deformations: the red parts indicate areas with the greatest deformation, and the blue parts indicate areas with the lowest deformation. The deformation vector field visualisations show the directions of these deformations.

In Fig. 3, it can be seen that M'Kali preshaped her trunk tip differently according to the type of food item she reached to grasp. As the Mahalanobis distances show, her trunk tip shape was similar when grasping apple slices and small cubes but different when grasping carrots or medium cubes. For the apple slices and small cubes, the deformations (indicated in red) were mostly on the sides of the trunk tip, and the vectors on these sides point toward the centre of the trunk tip. Thus, when grasping apple slices and small cubes, M'Kali's trunk tip was more elongated than usual. More specifically, when grasping apple slices, her trunk tip was more oval, and the top of the trunk tip was thicker; the shape of her trunk tip was similar when grasping small cubes but narrower in area. When grasping carrots, her trunk tip was more circular than usual, with a sharp top to the trunk tip, as indicated by the vector directions and isoline colours. Finally, when grasping medium cubes, M'Kali's trunk tip shape isoline visualisation was mainly blue. This trunk tip shape was similar to the usual shape, with only three small areas of difference, creating a slightly more rounded shape.

Figure 4 shows how the distal part of Juba's trunk was preshaped according to the type of food item grasped. Her trunk tip shape was similar when grasping small cubes and medium cubes but different when grasping carrots and apple slices. Moreover, as the Mahalanobis distances show, her trunk tip shape completely differed from the others shapes when grasping large cubes. As the deformation vector field and isoline visualizations for the

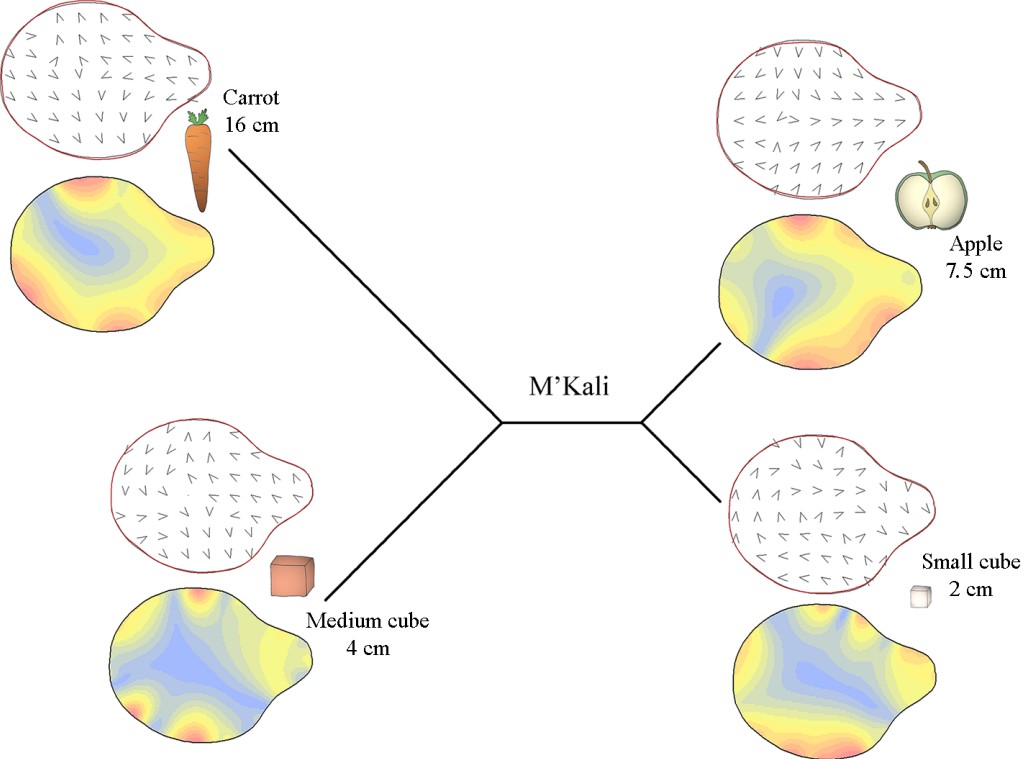

**Figure 3   Mahalanobis distance plots paired with thin plate splines deformation visualisations (vectors field deformations and isolines deformations) of trunk tip according to the object grasp, for M'Kali.** These Mahalanobis distance plots show through discriminant space the similarity between the different distal shapes obtained per object. The Thin Plate Splines deformations specify where and how the differences in form are on the distal shapes. Red parts on the isolines deformations reveal where the distal shapes change the most, and the blues parts where it change the least. Arrows on the vectors field deformations show in which direction the deformations occur.

periphery of the distal trunk shape show, her distal trunk shape became more circular than usual and the top of the trunk tip became thinner. When grasping other food items, the top of the trunk tip was thicker. Still according to the vector field and the isoline colours, when grasping small and medium cubes, Juba's trunk tip shape was slightly more circular than usual, especially on the left side for small cubes and the right side for medium cubes. When grasping carrots, her trunk tip shape was close to usual but thicker on the top right side of the trunk tip. Finally, when grasping apple slices, the general shape of the trunk tip was more oval than usual.

Figure 5 displays how Marjorie's trunk tip preshape differed according to the type of food item. Similar to M'Kali, her trunk tip shape was similar when grasping apple slices and small cubes but different when grasping carrots, medium cubes and large cubes. From the vector field and isoline colours of each object's distal trunk shape, we can see that when grasping apple slices and small cubes, Marjorie's distal trunk shape was more oval than usual, with differences on the left and bottom right of the trunk tip. Moreover, the top of the trunk tip was thicker than usual, especially on the left side. When grasping the three

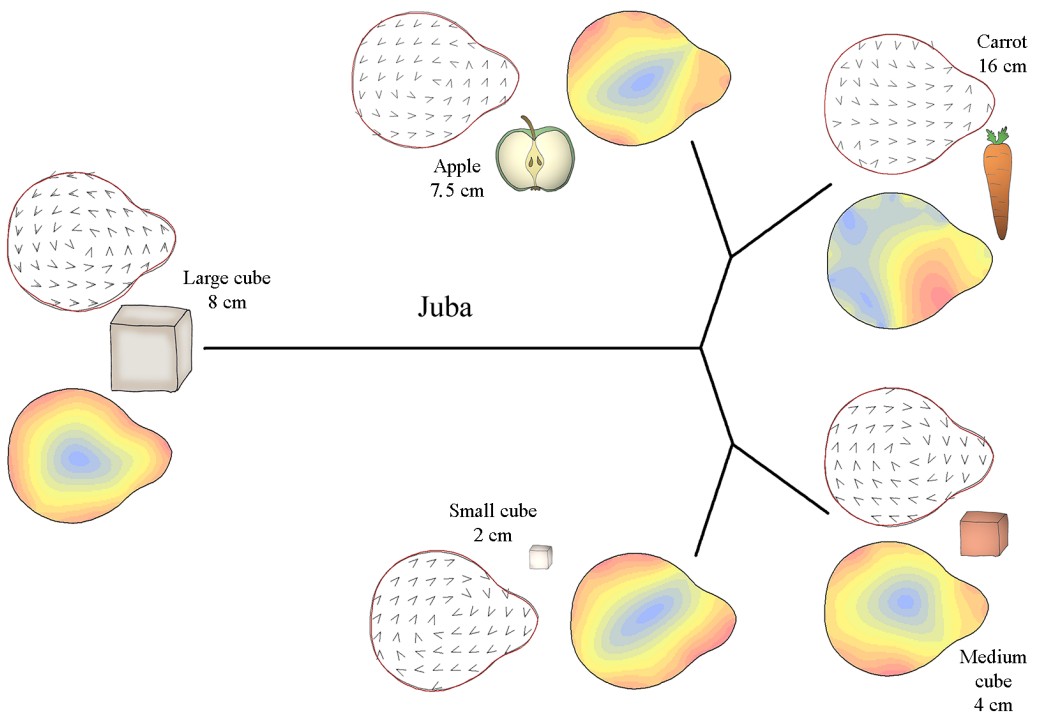

**Figure 4** **Mahalanobis distance plots paired with thin plate splines deformation visualisations (vectors field deformations and isolines deformations) of trunk tip according to the object grasp, for Juba.** These Mahalanobis distance plots show through discriminant space the similarity between the different distal shapes obtained per object. The Thin Plate Splines deformations specify where and how the differences in form are on the distal shapes. Red parts on the isolines deformations reveal where the distal shapes change the most, and the blues parts where it change the least. Arrows on the vectors field deformations show in which direction the deformations occur.

other types of food (carrots, medium cubes and large cubes), the top of Marjorie's trunk tip was thinner. The distal trunk shape was also more rounded than usual, especially on the right side when grasping medium cubes, on the bottom right side when grasping large cubes, and on the left side when grasping carrots.

Thus, M'Kali and Marjorie had more circular and thin-topped trunk tips when grasping carrots and medium cubes, while the top of the trunk tip was thicker for Juba. When grasping apple slices, the three elephants had more oval and thick-topped trunk tips. For M'Kali and Marjorie, the trunk tip shape when grasping small cubes was similar to that when grasping apple slices, but for Juba, the trunk tip shape was more circular. Finally, Marjorie and Juba had the same distal trunk shape when grasping large cubes: circular with a thin top.

The trunk tip shapes differed according to the type of object and individual, but the three elephants had many similarities in distal trunk shape for grasping the same object. M'Kali and Marjorie had the same trunk tip shapes when grasping the same objects. Juba tended to have the same shape as well with some food items, but her distal trunk shape also differed for other items, such as small cubes. Taken together, these results indicate that

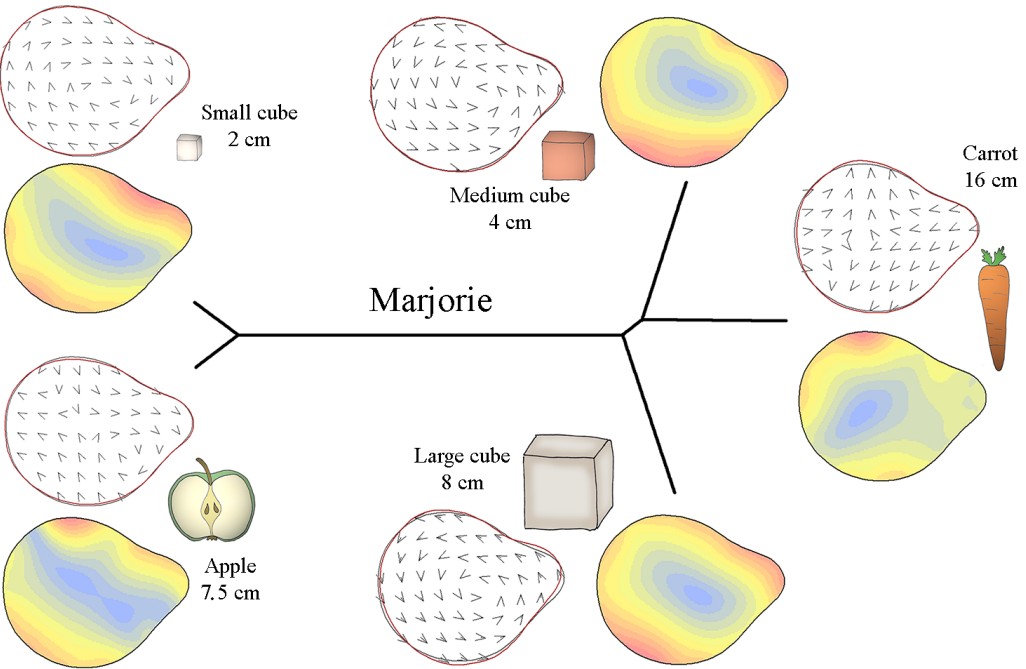

**Figure 5 Mahalanobis distance plots paired with thin plate splines deformation visualisations (vectors field deformations and isolines deformations) of trunk tip according to the object grasp, for Marjorie.** These Mahalanobis distance plots show through discriminant space the similarity between the different distal shapes obtained per object. The Thin Plate Splines deformations specify where and how the differences in form are on the distal shapes. Red parts on the isolines deformations reveal where the distal shapes change the most, and the blues parts where it change the least. Arrows on the vectors field deformations show in which direction the deformations occur.

there are both similarities and individual differences in trunk tip preshapes in the three elephants.

The distal part of the trunk was also preshaped in a way that matched the type of food item in these three elephants. A rounder trunk tip allowed it to better conform to the contours of food items such as carrots and medium and large cubes, with the sides of the trunk tip matching the edges of the items. Small cubes and flat apple slices, which were more difficult to grasp, required a more oval distal trunk shape to apply maximal surface area to the item. The more difficult the object was to grasp, the more oval the trunk tip shape was. Conversely, if the object offered more grip, as is the case for carrots, the distal trunk shape was closer to the shape of the object.

## Grasping success

Table 4 shows that trunk tip shape did not differ according to the number of failures before a successful grasp, namely, when the trunk hit the box but missed the food. There was no link between the distal trunk preshape and the number of failures for these individuals. Regardless of the number of failures before a successful grasp, the trunk tip was preshaped just before a grasp.

**Table 6  Mean of number of failures per individuals and per objects for all the grasping attempts.** The mean number of failure before a successful grasp, for each individual and for each object, on all grasping attempts, were calculated and are presented in this table.

| Variables | | Number of failures: mean |
|---|---|---|
| Individual | Ashanti | 0.98 |
| | Juba | 1.10 |
| | Tana | 1.18 |
| | Marjorie | 1.27 |
| | N'Dala | 1.43 |
| | M'Kali | 1.91 |
| Object | Large cube | 0.82 |
| | Carrot | 0.89 |
| | Medium cube | 1.24 |
| | Apple slice | 1.54 |
| | Small cube | 1.79 |

Moreover, Table 6 represents the mean number of failures per individual and per object for all the grasping attempts. Focusing on the individuals, the elephants that failed the most before a successful grasp were M'Kali and N'Dala, while Ashanti and Juba had the fewest failures. The three elephants that had the best grasping success (Ashanti, Juba and Tana) were not necessarily the same elephants that preshaped their trunk tip according to the object properties (M'Kali, Juba and Marjorie). Additionally, for these individuals, a distal trunk preshape in accordance with the object properties was not linked with the success of a grasping attempt.

The objects section of Table 6 displays the success of grasping attempts in general, including the mean number of failures per object for all grasping attempts. Failure was less likely when the elephants attempted to grasp large cubes and carrots and more likely when they attempted to grasp apple slices and small cubes. Overall, grasps were more efficient for large and thick food items.

However, over time (and with an increasing number of trials), the number of grasping failures consistently increased or decreased depending on the individual and objects concerned. Figure 6 shows the change in the number of failures on different grasping attempts for each individual and object pair. M'Kali and Ashanti had fewer failures with all food items as the trials progressed, notably improving their success of grasping small cubes (M'Kali) and carrots (Ashanti). At the beginning of the experiment, M'Kali had the most failures in grasping the small cubes of all elephants studied, but her number of failures quickly decreased as the trials progressed. Tana, Juba, Marjorie and N'Dala had fewer failures for most food items as the trials progressed, except for grasping carrots (Tana), apple slices and large cubes (Juba), apple slices and carrots (Marjorie), and apple slices and medium cubes (N'Dala). In particular, Juba had more failures in grasping the large cubes and fewer failures in grasping carrots as the trials progressed. Marjorie had the largest decrease in failures when grasping medium cubes, while N'Dala had the largest increase in

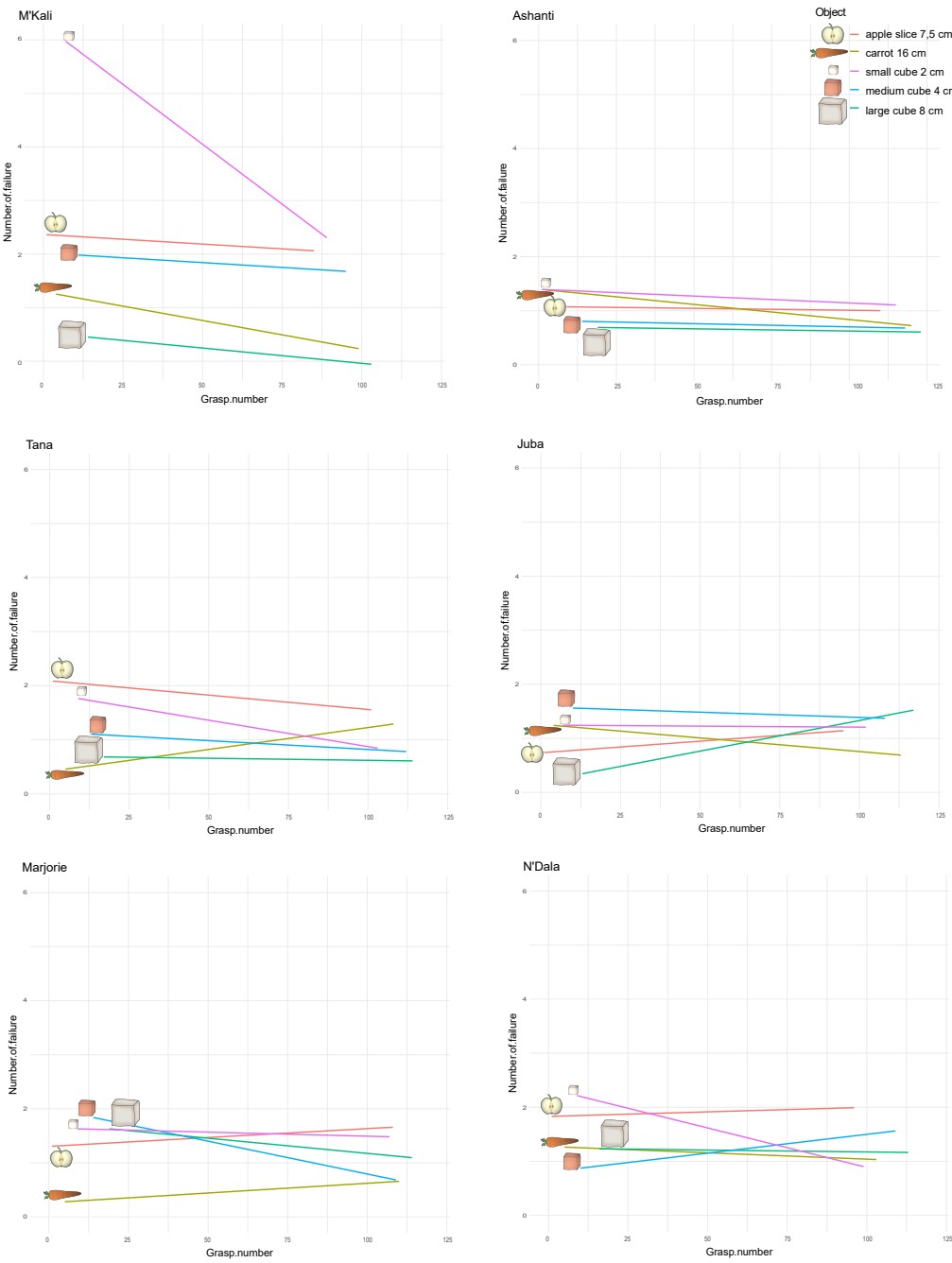

**Figure 6** **Number of failures along the grasping per individual and object.** The *x*-axes correspond to all the grasping perform over time by an individual, with all the objects. As we introduce the objects to the elephants one after the other each week, each object first grasp number then depends on when the elephant was first introduce to the object in its attempts. The *y*-axes represent the number of failure before a successful grasp, namely when the trunk hit the box but missed the food. The straight lines show these number of failure tendency along the attempts and per object, for each individual.

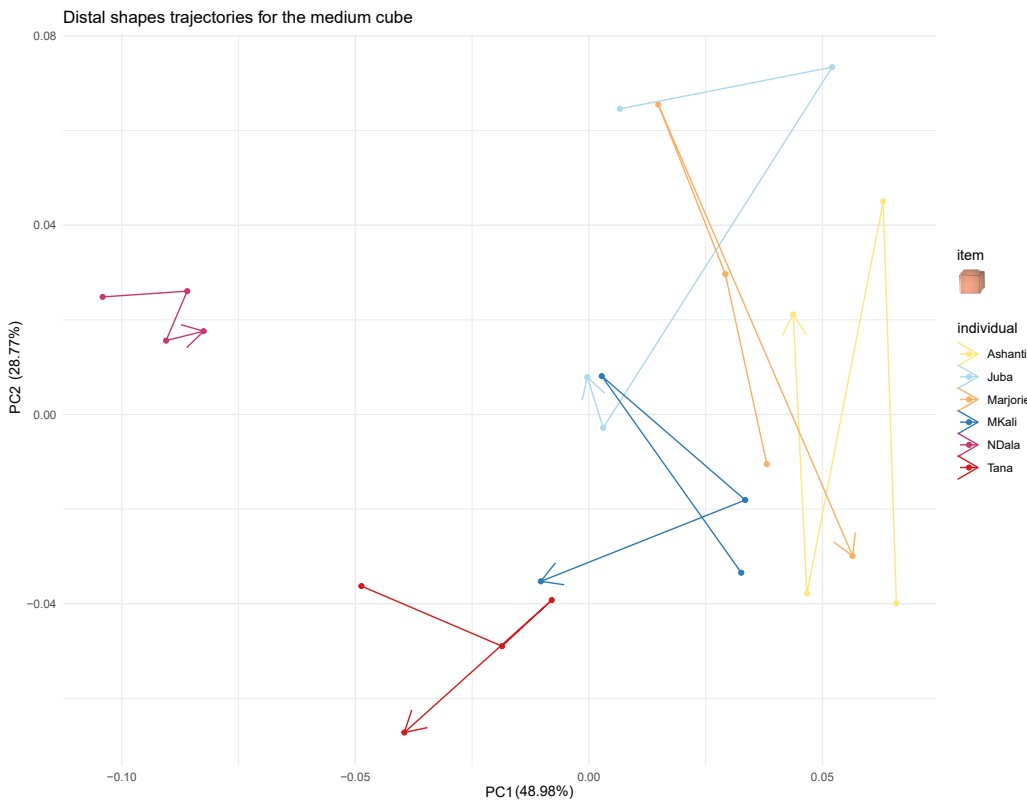

**Figure 7** **Distal shapes trajectories of all the individuals when grasping a medium cube.**

failures when grasping medium cubes and the largest decrease in failures when grasping small cubes.

All the studied elephants improved in grasping efficiency over time, with all or the majority of the food items. However, each elephant improved on different objects.

Finally, we investigated whether the trunk tip shapes of all the individuals converged when grasping the same object by examining the trajectories of distal trunk shapes over time. Figure 7 displays these trajectories for all studied elephants when grasping medium cubes. The trajectories when grasping the other types of food items are provided in the Supplementary Material. In Fig. 7, each individual's distal trunk shape when grasping medium cubes changed over time. However, no individual's preshapes clearly trended in the same direction. Thus, their trajectories when grasping medium cubes did not converge. The results were similar for the trajectories when grasping other food items. Taken together, these results did not indicate a convergence of individual distal trunk shapes when elephants grasped the same type of object.

## DISCUSSION

We found that elephants preshaped their trunk tip in reach-to-grasp movements toward food items according to the characteristics of the different food items. Preshaping varied

between individuals, which each had a unique distal trunk shape, but was not linked to grasping success.

## Individual differences

As shown by our results, all the elephants in our study had a unique trunk tip shape. Similar results are found in the human literature: individual and interpopulation variation in human hand forms are well documented (*Králík, Katina & Urbanová, 2014*) and have a real impact on hand utilisation and related abilities. For example, some sports modify and intensify these individual differences in hand shape, which confers an advantage for certain hand shapes in these disciplines (*Barut, Demirel & Kıran, 2008*; *Fallahi & Jadidian, 2011*). Thus, we assume the same applies to elephants: different trunk tip shapes may lead to different reach and grasping skills in individuals. These individual differences may influence (and predict) individual grasping performances.

## Trunk tip shapes and object characteristics

We highlighted a significant effect of food item properties on the trunk tip preshape of three out of the six elephants. M'Kali, Juba and Marjorie adjusted the shape of their trunk tip according to the object properties, consistent with findings of preshaping in the hands or paws of other species (*Chieffi & Gentilucci, 1993*; *Christel & Fragaszy, 2000*; *Roy et al., 2000*; *Sacrey, Alaverdashvili & Whishaw, 2009*; *Sartori et al., 2013*). These elephants also showed similar distal trunk preshapes for almost all the types of food items in our experiment; the preshapes matched the shape of the food and its difficulty to grasp. This adaptation of trunk tip shape may allow individuals to grasp more efficiently. Comparable results are found in studies on hand or paw preshaping according to object properties (*Christel, 1993*; *Pouydebat et al., 2011*). In animals such as primates, preshaping sometimes provides more appropriate and optimal grasping (*Pellegrino, Klatzky & McCloskey, 1989*; *Christel, 1993*; *Supuk, Kodek & Bajd, 2005*; *Pouydebat et al., 2009*).

Interestingly, preshaping of the trunk tip according to the object properties does not occur in all elephants, as half of our sample did not show any trunk tip shape adaptation. Although the preshaping of hands or paws is found in many primates and rats (*Napier, 1993*; *Sacrey & Whishaw, 2012*), it is not always widespread in all subjects. Preshaping can result from individual differences in hand, paw or trunk tip morphology but may also be the result of individual variability in motor control.

Moreover, the group of elephants that did not preshape their trunk according to the food item properties included the blind elephant. According to Jeannerod, preshaping is controlled by visuomotor channels and, more precisely, by visual data of the object shape (*Jeannerod, 1981*; *Jeannerod, 1988*). *Weir, (1994)* corroborated this idea. In humans, a grasping movement without visual feedback will have an increased hand aperture (*Uno et al., 1993*; *Winges, Weber & Santello, 2003*; *Tipper, Paul & Hayes, 2006*; *Sacrey & Whishaw, 2012*). Although *Santello, Flanders & Soechting (2002)* showed that continuous visual feedback from the object or the hand is not necessary for adequate preshaping, a degree of visual input remains essential. Some primates and other animals cannot preshape their hands or paws because of vision impairments (*Uno et al., 1993*; *Winges, Weber & Santello, 2003*;

*Tipper, Paul & Hayes, 2006*). This may be the case for the blind elephant, N'Dala, that did not exhibit trunk tip preshaping in accordance with the properties of the object grasped. To verify this hypothesis, the experiment could be redone with entirely blind grasping tests for all subjects. However, like the other elephants, N'Dala still managed to grasp all the food items. These results confirms the idea that vision, one of elephants' less-used senses (*O'Connell-Rodwell, 2007*; *Poole & Granli, 2011*; *Plotnik et al., 2013*), is not necessary for grasping food items with the trunk.

The subadult elephant, Ashanti, also did not exhibit trunk tip preshaping according to the properties of the food item. This may be due to her relatively young age and interactions with other elephants; in some species young and subadult individuals do not exhibit preshaping or use adult grasping strategies (*Forssberg et al., 1991*; *Butterworth & Itakura, 1998*; *Pouydebat et al., 2011*; *Le Brazidec et al., 2017*). Thus, preshaping of the trunk tip according to the object properties may result from learning.

## Grasping success

We did not find a link between the trunk tip shape just before a grasp and the number of failures before a successful grasp. Moreover, unlike what we hypothesised and what is found in other species (*Christel, 1993*; *Supuk, Kodek & Bajd, 2005*), preshaping the trunk tip did not increase grasping success. Therefore, trunk tip preshaping may not confer a selective advantage in terms of reducing the number of failures before a successful grasp for these elephants.

Although grasping is logically more efficient with large and thick food items, all the studied elephants improved in grasping efficiency over time, each with different objects. This result suggests that each individual developed, over the trials, her own grasping strategies to increase grasping success. This individuality of grasping success for individual–object pairs might indicate taste preferences. Individuals may be motivated to improve grasping success for foods that they prefer and may be less motivated when they find the food less appealing. These differences in grasping success can also be explained by the constraints of the experiment: we could not control the posture of the elephant in front of the box or the position of the head relative to the bars. The elephants may have experienced different visual obstructions while grasping the food items; however, N'Dala, the blind elephant, did not have significantly lower grasping success than two of the elephants that preshaped their trunk tip according to the object properties (Marjorie and Juba). Thus, vision may have little importance to grasping success of specific objects by individuals. As olfaction is essential in elephant life (*Polla, Grueter & Smith, 2018*; *Plotnik et al., 2013*; *Plotnik et al., 2019*), we assume that elephant trunk grasping success depends more on olfaction than on vision.

Finally, in contrast to our hypothesis, individuals' distal trunk shapes did not converge when grasping the same object. Although the three individuals who preshaped their trunk tip according to the object properties tended to have the same general shape when grasping the same type of object (more oval or more rounded), over time, they did not develop similar grasping shapes. This was also true for the three other elephants. In the studied

elephants, there was no general convergence of trunk tip shape over time to optimise grasping success.

## CONCLUSIONS

To conclude, our results indicate that three out of the six studied African savannah elephants adjusted their distal trunk shape to that of the object. Preshaping varied depending on the individual, each of whom had a unique distal trunk shape. This study is the first time that preshaping has been demonstrated with a muscular hydrostat, in other words, with a grasping appendage that has no joints. Grasping strategies can be similar even if the grasping appendage operates in a completely different way. We found that the blind elephant could grasp but did not optimally preshape its trunk. This finding reveals that elephant trunk grasping success may depend more on olfaction than on vision, but preshaping during reach-to-grasp movements is probably, as in other species, mostly controlled by visuomotor channels. Moreover, as the blind elephant had a similar increase in success grasping objects as the other elephants, we assumed that grasping success did not depend only on vision. Preshaping may also result from learning, as one of the other elephants that did not exhibit optimal preshaping was a subadult. The third elephant in this group, Tana, did not show an optimal preshaping. This may be due to her life history, such as experience or learning during development. It would be interesting to investigate further, perhaps by conducting a developmental preshaping study. However, preshaping of the trunk tip did not increase grasping success and thus may not confer a selective advantage in these elephants. We observed increases in grasping success that differed according to objects and individuals, probably influenced by learning and food taste preferences.

Finally, our study was based on a sample of six African savannah elephants. It would be interesting to increase the sample size and to carry out a similar study in Asian elephants, which have only one finger on their trunk, to understand preshaping strategies in elephants more broadly.

## ACKNOWLEDGEMENTS

We would like to thank ZooParc of Beauval (France), which enabled us to conduct this experiment with their elephants. The experiments were conducted with assistance from the keepers Amaury Boutier, Nathan Durand, Matthieu Fromet, Mathieu Hysbergue, Clément Langles, Mégane Marron and Matthieu Villemain and under the direction of Yann Ménager. We warmly thank the entire team. We also thank Arnaud Delapré for his technical help. Finally, we would like to thank our two reviewers, Andrew Schulz and the anonymous reviewer, for their helpful comments, which greatly improved our paper.

### Funding
This work was supported by the funding CNRS 80 PRIME (E. Pouydebat). The funders had no role in study design, data collection and analysis, decision to publish, or preparation of the manuscript.

### Grant Disclosures
The following grant information was disclosed by the authors:
CNRS 80 PRIME.

### Competing Interests
The authors declare there are no competing interests.

### Author Contributions
- Julie Soppelsa conceived and designed the experiments, performed the experiments, analyzed the data, prepared figures and/or tables, authored or reviewed drafts of the paper, and approved the final draft.
- Emmanuelle Pouydebat conceived and designed the experiments, prepared figures and/or tables, authored or reviewed drafts of the paper, and approved the final draft.
- Maëlle Lefeuvre conceived and designed the experiments, performed the experiments, authored or reviewed drafts of the paper, and approved the final draft.
- Baptiste Mulot and Celine Houssin conceived and designed the experiments, authored or reviewed drafts of the paper, and approved the final draft.
- Raphaël Cornette conceived and designed the experiments, analyzed the data, prepared figures and/or tables, authored or reviewed drafts of the paper, and approved the final draft.

### Animal Ethics
The following information was supplied relating to ethical approvals (i.e., approving body and any reference numbers):

We followed the French ethical standards to work in a zoo without any interventions on the animals. All methods were performed in accordance with the relevant CNRS guidelines and the European Union regulations (Directive 2010/63/EU).

### Data Availability
The codes (in R) and raw data are available in the Supplementary Files.

### Supplemental Information
Supplemental information for this article can be found online at http://dx.doi.org/10.7717/peerj.13108#supplemental-information.

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
