# Peer review of "The relationship between distal trunk morphology and object grasping in the African savannah elephant (Loxodonta africana)"

_PeerJ, doi:10.7717/peerj.13108_

## Round 0.1 · original submission · Major Revisions

Dear authors,

Thank you for your submission to PeerJ. I very much enjoyed this paper and think that it will be a useful addition to the literature pending major revisions based on comments from two reviewers. Reviewer 1 notes issues with the figures (add scale bars and increase the size of some of the smaller panels). Both reviewers also commented that the figures should incorporate the images of the food items more (similar to figure 3, which is very effective). Figure 4 is particularly difficult to interpret and could be quickly improved. Further, both reviewers note many grammatical issues present throughout the paper that need to be fixed prior to publication. These issues often reduce clarity and distract from the point that is being made. Having a proficient English speaker read through the paper would be very beneficial.

When submitting your revisions, please include a clean version of the manuscript, a tracked changes version, and an itemized document going through each reviewer comment.

Please let me know if you have any questions.

Best,

Brandon P. Hedrick, Ph.D.

·

Basic reporting

Overall the manuscript is well written in its preparation and organization of scientific data. I think the authors could benefit from reading some of the sentences out loud as some of the sentence structure and use of passive versus active voice can shift. Overall the background needs improvement with more sources tying in the similarities and differences between the distal tip of the African elephant versus the tip of Asian elephants as well as other primate species.

The largest improvement in the manuscript would be on the figures. Many of the figures contain inconsistancies and I believe adding some figures/tables labels and schematics could greatly increase the context of this article.

The authors have a clear knowledge and background of the idea of reaching tasks, but I would like to see some more tie in to the unique morphology of the trunk and comparisons to more muscular hydrostats.

Experimental design

Personally the experimental design seemed sufficient towards understanding the author's hypothesis. I would prefer for the authors to include more detailed schematics in the figure to help the readers understand what the actual methodology of this project is. I give more information in the line by line comment, but I would recommend not using each of the elephant's names and completely dismissing the elephant that was used in analysis on only some points. The authors list that the experiments were performed on 6 elephants, however 5 elephants were only used in some of the analysis. In this field your sample size could be indicated to be not-sufficient, but working with elephants is difficult and to me a sample size of 5 is still rigorous for much of your analysis.

Validity of the findings

I would like to see more discussion again to compare what the authors might think in the Asian elephant as the bottom finger on the trunk is composed of cartilidge. The authors also simplified this down to 2D and I'm curious if there is any thought given to how the trunk movement before the tip is changing. The conclusion itself is well stated, but I think targeting a more morphological argument is needed as many of the comparisons given are to animals that have joints that they shift and there is no discussion of joints with the elephant as it is purely a muscular hydrostat.

Additional comments

Line 31 – At the beginning of the abstract it would help if you indicated that you are talking about the hand of a human. There are even descriptors of elephant trunks being called the “hand of the beast” that you could use in your abstract as a transition.
General Abstract – I think it could benefit from discussing more about Muscular Hydrostats in general. Look to many of Bill/William Kier’s publications on discussion of muscular hydrostats as tongues, trunks, and tentacles and you have a clearer background of comparison. Also in the abstract it would help to find a more similar morphology comparison than that of the hand to the reader.
Line 40 – With the new updated African forest elephant species in recent years. Most literature I see cites African bush elephants as (Loxodonta africana)
Line 53 – social interaction, you could discuss as this relates to elephants by citing that elephants often grasp eachother’s trunk’s in a trunk shake type of movement.
General Introduction Comments – Something you should bring up in the introduction is the idea that African elephants which you studied have two fingers at the tip of their trunk, where as Asian elephants only have one. Your method of grasping behaviors is relevant to many of the primate species listed as they also have thumbs which allow them to grip objects in a different way. As African elephants are browsers and Asian elephants are grazers there are different evolutionary reasons why they developed differences at the tip of their trunks.
I recommend adding some of the more recent citations pertaining to elephant trunk reaching movement including: https://www.sciencedirect.com/science/article/pii/S0960982221011337 , https://royalsocietypublishing.org/doi/10.1098/rsif.2021.0215
Line 122 – I would recommend shying away with naming the elephants in the study and instead identifying them by Elephant A, B, C, D, etc. or Elephant 1, 2, 3, 4. This is good to not indicate specific challenges individual elephants have for ethical reasons. I would encourage you to also include a table giving the age, sex, weight, etc. of each of the elephants to allow readers to compare/contrast differences in their gross characteristics.
Line 140 – The paper’s methods could be more useful to future studies if exact descriptors of the quantities were described as well as exact dimensions. Perhaps including a table of all of the exact dimensions of the food items could help with understanding length scale differences.
Line 162 – was any training done with the elephants to perform this task. If so how long did the training last and did you wait until a certain percentage for the elephants to successfully reach.
Line 217 – I’m a bit confused about how the 107 number was found. Did some of the individual elephants reach more than others or was it completely consistent. Especially with you not including the data from N’Dala. In my opinion if you are excluding N’Dala from one part of the analysis it would be best to just exclude those trials all together and not only include that data when applicable.
Line 221 – you indicate that the p-value is significant but what hypothesis did this prove or disprove?
Line 225 – its important to know that just because the p-value is significant does not indicate that they do not differ. It just indicates that the hypothesis test failed. Therefore you cannot conclude that opposite of a failed hypothesis test.
General Figure comments – every one of the figures need scale bars to indicate the size of many of the images. Especially for the experimental design figure 1. A lot of your figures are very small. It is very difficult to read what exactly the plots are saying especially you’re A, B, C.
Figure 4 – you use different graphics of carrots, apple slices etc. on previous graphs, I would recommend adding those to the figure 4 so readers know what each line is. I really think Figure 4 could be visualized better in another form. Each one of your sub-graphs has a different x and y axis in terms of your figures are labeled so some of the differences seem a lot larger than they actually are. You should graph all of them on the same axis with a grasp number from 0-120 and a number of failure from 0 to 6 as those are the entire ranges of your experimental design.

Reviewer 2 ·

Basic reporting

Some issues with the clarity of the writing. I have provided more details below - I have attempted to provide examples for how certain sections could be re-written.

Experimental design

Experimental design is clearly laid out.

Validity of the findings

Methodology and statistics appear correct.

Additional comments

This study assesses how elephants will ‘pre-shape’ the distal aspect of their trunks in preparation for picking up an item, such as some food. The authors examine this behavior in six elephants and find that elephants do indeed pre-shape their trunks depending on food size/shape. They go on to suggest that the ability to pre-shape is likely a learned behavior and is based on visual cues (given that a sub-adult and a blind individual exhibit a much lower instance of pre-shaping).
I find the methodology very compelling and the analyses well designed. I can appreciate that large animal biology is difficult to conduct, and there are many logistical challenges associated with getting access to elephants. I feel the results nicely extend the thinking of animals use their appendages for various tasks.
I have a handful of (mostly minor) comments I hope the authors can address. I also offer some help with re-framing certain sentences.

# # #

Abstract
L35 Appendices – do you mean appendages? There are a couple of other places where this appears in the manuscript (i.e., L80, L99).
L46-48 I think there’s a missed opportunity to quickly explain why you only have 3 individuals conforming to your hypotheses about pre-shaping. When I originally saw that only 3/6 elephants pre-shape their trunks I was quite surprised – but you should mention that those that did not were either blind, or sub-adults, indicating that there is likely some learning occurring, and that visual cues are important.

Introduction
L71 ‘when the hand moves forward the object’ -> ‘when the hand moves toward the object’
L105-107 Please re-phrase. I think you’re asking if all elephants exhibit similar trunk grasping shapes for a given item.

Methods
General: You should mention that the blind and sub-adult elephants are used to assess the role of vision and experience respectively in grasping behaviour. This better prepares are reader fop what’s coming.
L164 ‘We captured these prints at every first put of the trunk tip on the box and…’ -> ‘We captured these images when the elephant first approached the box and…’
L166-167 Is this the only definition of a feeding failure? I think this needs a more comprehensive definition. My original thought was maybe a failure occurs when the elephant drops the food item, or some set amount of time elapses as the elephant can’t pick up the item. Just knocking the box seems a little harsh.
L172 ‘print’ -> image
L173 ‘posed’ -> placed
L200 ‘grille’ -> grid
L209 How were these trajectory plots performed. Are these outputs from the trajectory.analysis() R function in geomorph – if yes, some statistics should be reported. For example: magnitude (P.magnitude.diff), angles (P.angle.diff), and shape differences (P.shape.diff). All these describe slightly different aspects of the trajectory that should prove your point. These, and perhaps other outputs depending you your goal, should be reported in the text or in a table.
L209-211 Bit lost here. Please rephrase – I think you’re trying to say the following. ‘These trajectories of the distal shapes over the grasping permitted to see a potential merging…’ -> ‘Trajectories of the distal trunk during grasping often exhibited similar shapes among individuals for the same object.’

Results
L257 ‘the deformation is the most, and the blue parts where the deformation is the less.’ –> ‘the deformation is greatest, and the blue parts are where the deformation is lowest.’
L262 typo
L277 ‘became more thicken.’ -> ‘became thicker.’
L308 ‘The most the object’ -> ‘The more the object’
L319 ‘when Ashanti and Juba mad the’ -> ‘while Ashanti and Juba made the’
L320-322 I don’t understand this sentence. Are you saying that all elephants had a comparable number of failures?
L330 ‘The figure 4 shows’ -> ‘Figure 4 shows’
L346 The figure 5 displays’ -> ‘Figure 5 displays’

Discussion
L366 typo
L387 ‘the preshaping of the hand or paw can be find usually in most of individuals in primates and rats’ -> ‘the preshaping of the hand or paw can usually be found in many primates and rats’
L398 ‘the vision stays primordial’ Please rephrase, I don’t know what this means.
L434 though -> thought

Conclusions
Any thoughts as to why Tana did not preshape the trunk?
L445 grasped -> grasp

Figures
Figure 2 You could also code the point symbols to reflect each food item (i.e., circles for the carrot, squares for the small cube, diamonds for the large cube etc.)
Figure 3 (very nice figure) Was M’Kali never given a large cube? Based on the supplementary trajectory figures it appears M’Kali rarely interacted with the large cube, you should mention so in the methods or results.

---

## Round 0.2 · Minor Revisions

Dear authors,

Thank you for your submission to PeerJ. I believe that your paper will be publishable in PeerJ following some minor revisions, primarily surrounding issues with the language. There are many grammatical errors throughout the paper and the paper cannot be published until they are remedied. In addition, reviewer 1 has some suggestions to improve the figures, which should be implemented.

I went through the first half of the paper and corrected many of the grammar issues (see below). However, please have a native English speaker fully revise the paper before resubmitting it. I see in your cover letter that you had a proficient English speaker go through it, but there are still many, many errors.

Please let me know if you have any questions going forward and I will be happy to answer them. Thank you again for your submission.

Best,

Brandon P. Hedrick, Ph.D.


Line 31–32: “depends on the food’s properties”

Line 32–33: I’m not sure what this sentence means.

Line 35–36: ‘that grasp objects’

Line 36: ‘called a hand due to its impressive’

Line 37: ‘few data have been collected on’

Line 40: ‘trunk during grasping’

Line 42: ‘flat’

Line 44: geometric morphometric should not be capitalized

Line 59: ‘during greeting ceremonies, bonding ceremonies, and during social play’

Line 68–69: Change to ‘Many primates preform their hands according to food properties during grasping, including chimpanzees…’

Line 72: ‘human’s’

Line 75: Maybe just ‘grasping strategies’?

Line 83: ‘enables increased grasping success’

Line 96: ‘congeners or other items in their environment’

Line 98: ‘like cephalopod tentacles’

Line 98: ‘and can bend in any direction, elongate, shorten, and lift heavy masses’

Line 101–102: ‘Elephant grasping is also increasingly studied’

Line 104: ‘can also be found in’

Line 111: Reword this sentence. Not sure what you mean by morphology and capacity.

Line 112: ‘trunk tips are then sometimes compared’

Line 113: ‘like the human hand’

Line 114: ‘Despite these’

Line 118: ‘inter-individual’

Line 124: reword ‘distal part shape’

Line 148: ‘used to grasping food’

Line 149: What do you mean by ‘thus no practices upstream the experiment were necessary’? Reword.

Line 161: plural for celery is celery

Line 177: ‘inlayed’

Line 178: ‘where the items’

Line 179: ‘maintained’

Line 180: ‘below’

Line 183: ‘GoPro 7’

Line 186: ‘for each grasp’

Line 193: do not capitalize geometric morphometrics

Line 195: ‘obtained’

Line 197: Trunk tip instead of top?

Line 198: ‘trunk distal contours’

Line 199: ‘characteristic’

Line 200: ‘annotated’

Line 203: Capitalize ‘Procrustes’

Line 209: Capitalize ‘MANOVA’ throughout

Line 212: ‘Variates’

Line 216: ‘deformed shape’

Line 217: ‘All the other analyses included all six elephants’

Line 219: I think you mean to calculate Mahalonobis distances between food items

Line 222: ‘plotted’

Line 236: ‘Procrustes’

Line 236: ‘first and last’

Line 246: ‘is not’ rather than ‘isn’t’

Line 248: ‘for each session’

Line 248: ‘The low number of graspings for the large cube is also explained by the…’

Line 249: ‘only interacted’

Line 251: ‘occurred in the 148 graspings that were removed’

Line 256: ‘support the hypothesis’. You do not prove hypotheses. Change this throughout.

Line 259: ‘trunk tip shapes’. Also fix this on line 260

Line 268: ‘during food grasping is shown’

Line 269: ‘grasping’

Line 279: ‘shows that there are different trunk tip shapes according’

Line 287: ‘figure 3 displays..’

Line 288: ‘interacted’

Line 289: ‘was not included’

·

Basic reporting

Overall it appears the authors did an excellent job with the reviews. I really appreciate the re-organizing of the background and it flows really well together with the rest of the paper. I appreciate the author's improvement of the figures.
I have included additional comments on the figures that once made I believe the manuscript is ready for publication.
One item that would still help is by changing the color and size of the scale bar based on the background imagery. So for example in Figure 1C, making the text & scale bar white font. Also updating the thickness of the bar would be beneficial.
I believe Figure 2 should still be improved in terms of making sure each of the axes are scaled the same way. Currently, the CV1 axis is skewed to the left. Making sure the axis positive and negative labels are equivalent will help not bias the appearance of the data for readers.
Figure 3, 4, 5 I would add text labels next to “Apple 1 cm, Carrot 1 cm cubes, etc.” As there is no scale to indicate the number/size of items. This comment is as on M’Kali’s plot there is a small white cube, but on Juba’s plot, there is a small and large white cube. So distinguishing the difference between each of those would help.
Figure 6 would be improved by clarifying the large cube/medium cube/small cube by size. Indicating on the figure each of those would help the readers understand the flow from small to large in the differences.

Experimental design

Including one video of each of the elephants as well as reaches is really beneficial. One thing that could help the authors is by including a single supplemental video file that the authors can then cite in the manuscript in the methods section to help readers have a video to see the experiment live. Also I agree with the authors rebuttal of keeping the names of the elephants as with being more explicit with the number of individuals used for each analysis makes much more sense.

Validity of the findings

The authors made substantial improvements to the discussion and results sections following the reviews provided. There are additional arguments and discussion points made about the uniqueness of this study in terms of muscular hydrostats.

Additional comments

Additionally I would like to apologize to the authors for my delay in getting the review back to you. The authors clearly did quite a bit of work to improve the manuscript and with some additional figure edits I look forward to seeing this scientific work published.
Also I would like to commend the authors on including members of the Zoo where the research was conducted as co-authors. Historically many zoo-personnel are not included as authors and I appreciate you all doing good, collaborative, and well-meaning science.

Reviewer 2 ·

Basic reporting

Writing is much improved, but could do with one more run through.

Experimental design

Experimental design is clearly laid out.

Validity of the findings

Methodology and statistics appear correct.

Additional comments

This study assesses how elephants will ‘pre-shape’ the distal aspect of their trunks in preparation for picking up an item, such as some food. The authors examine this behavior in six elephants and find that elephants do indeed pre-shape their trunks depending on food size/shape. They go on to suggest that the ability to pre-shape is likely a learned behavior and is based on visual cues (given that a sub-adult and a blind individual exhibit a much lower instance of pre-shaping).
This is the second time I’ve seen this paper and some clear improvements have been made. I’m glad to see that the authors took the time to respond to my comments in the rebuttal letter – I agree on the CVA plot, the current version is superior.
I have no major comments and just a handful of minor comments that I think the authors should be able to easily incorporate. I would however ask the authors to have another person check the English again – it’s much improved, but there are some sentences that may benefit from a quick edit.

Minor Comments
L69 Typo? macacas -> macaques
L180 An example of grasping film can be found in the supplementary data – I couldn't see this among our files but I trust the final version will include this video.
L198 These landmarks formed the trunk distal parts contours, and then this 2D sliding depicted the different distal shapes. – I'm not really sure what you're trying to say here. Perhaps that the semi-landmarks defined the distal borders of the trunk tip?
L203 The GPA serves to remove the effects of translation, rotation, and scaling from your shape data – I'm not sure why you're talking about correlation here.
L236 typo? proscruste space -> Procrustes space

---

## Round 0.3 · Minor Revisions

Dear authors,

Thank you for your careful response to reviewer comments. The manuscript now reads very well and I think is quite interesting. On a final read through, I caught a few more grammar errors and just a few areas that should be clarified. Once they are done, I believe the manuscript will be publishable in PeerJ.

Please let me know if you have any questions going forward and I will be happy to answer them. Thank you again for your submission.

Best,

Brandon P. Hedrick, Ph.D.



Line 199: probably ‘tpsDIG2’?

Line 208: What version for geomorph and for R? Also on line 219 for morpho and Momocs on line 237.

Line 230: ‘showed the difference in shape’

Line 342: ‘carrots’

Line 502: But isn’t this in contrast to what you said about olfaction?

Table 3: ‘Number of grasping attempts per…’. Also in the table itself, change to ‘number of grasping attempts’ and ‘total grasping attempts’.

For tables 4 and 5, please report the full results of the MANOVA rather than just the p-values

Table 6 caption: ‘for all the grasping attempts’

Figure 1 caption: Do you mean for (B), ‘Three of the six African savannah elephants’?

Figure 3, 4, 5 captions: ‘thin plate spline deformation visualisations…’

Figure 3, 4, 5 captions: ‘least’ rather than ‘less’

Figure 3, 4, 5 captions: ‘show in which direction the deformations occur’

Table 6 and figure 6: I’m not sure what I’m looking at here. On the x-axis is grasping number starting with the first grasp and continuing. Is this the number of grasp attempts before a failure? So for M’Kali grasping a small cube, she failed 6 times before successfully grabbing it on the first grasp attempt? I’m also not sure why all of the grasp.numbers don’t start at 1. Can you go into this a bit more in the figure caption, table caption, and methods just to clarify for readers?

---

## Round 0.4 · accepted · Accept

Dear authors,

Thanks for your quick revisions. I noticed a few more things as I was reading through the corrections, but they can be fixed in the next stage I think.

Line 243–244: I’m not exactly sure what you mean here, but perhaps: ‘…the number of failures are shown for each…’ I’m not sure if this is what you mean and it might change the meaning. Also it looks like this is different between the tracked changes version and uploaded manuscript.

Line 246: Procrustes spelling

Line 505–506: ‘may depend more on’ rather than ‘depends’

Thank you again. Let me know if you have any more questions as the manuscript moves to the next stage.

Best,

Brandon P. Hedrick, Ph.D.